# Parallel Scheduled Sampling

## Abstract

Auto-regressive model architectures are the workhorse of sequence generation tasks in text, images, audio, and more. These models are predominantly trained with *teacher-forcing*, wherein the model is encouraged to predict the next ground-truth token conditioned on all previous ground-truth tokens in the sequence. At test time, the model does not have access to ground-truth tokens and must instead condition on its own prior predictions, resulting in a train-test discrepancy known as *exposure bias* (Ranzato et al., 2015). Scheduled Sampling (Bengio et al., 2015) aims to mitigate exposure bias by randomly replacing some tokens with model predictions during the course of training. Unfortunately, this technique is inherently sequential and unlike teacher-forcing, cannot be trivially parallelized across time. In this paper, we introduce a simple technique to parallelize Scheduled Sampling, significantly increasing training throughput. Experimentally, we find the proposed technique leads to equivalent or better performance on image generation, summarization, dialog generation, and translation compared to teacher-forcing. In dialog response generation, Parallel Scheduled Sampling achieves a 1.6 BLEU score (11.5%) improvement over teacher-forcing with more than 250 times higher throughput than Scheduled Sampling. In image generation, our approach achieves 20% and 13.8% improvement in Fréchet Inception Distance (FID) and Inception Score (IS) respectively. Finally, we discuss the effects of different hyper-parameters associated with Scheduled Sampling on the model performance.

## 1 Introduction

Auto-regressive models are a popular choice for generating sequences of any kind including audio (van den Oord et al., 2016b), images (van den Oord et al., 2016a), and text (Sutskever et al., 2014; Cho et al., 2014). Here, the joint probability of the sequence is factorized in a pre-determined order during train and test time. For example, auto-regressive models for text generation factorize the joint probability left-to-right. The text sequence is generated by a decoder network left-to-right, one token (word or word-piece or character) at a time and are widely used in text generation tasks such as summarization (Liu et al., 2018), machine translation (Sutskever et al., 2014) and dialog response generation (Budzianowski et al., 2018) in the encoder-decoder (Cho et al., 2014; Sutskever et al., 2014) setting. Such models are typically trained by *teacher-forcing* (Williams and Zipser, 1989) where ground-truth history is fed to the model as input, which at test time is replaced by the model prediction. Auto-regressive models applied in any domain suffer from this train-test time discrepancy.

Scheduled Sampling (Bengio et al., 2015) aims to mitigate the discrepancy between train and test time in teacher-forcing by randomly replacing some tokens in the history with the model's prediction. More concretely, at a given time step in generating the output sequence, the model is conditioned either on ground-truth or model prediction from the previous time-step with some probability. The probability of selecting model predicted token is gradually increased as training progresses. This procedure potentially allows the model to recover from its own errors, and Bengio et al. (2015) observe better empirical performance in natural language parsing, image captioning, and speech recognition compared to teacher-forced training. Scheduled Sampling has also been used to get better performance in other tasks such as video prediction (Finn et al., 2016), knowledge base completion (Miwa and Bansal, 2016) and piano music transcription (Sigtia et al., 2016).

A key bottleneck in training models with Scheduled Sampling is its inherently sequential nature. Unlike teacher-forcing, tokens must be processed one time-step at a time. The sequential procedure makes Scheduled Sampling impractical for training neural networks, particularly on problems

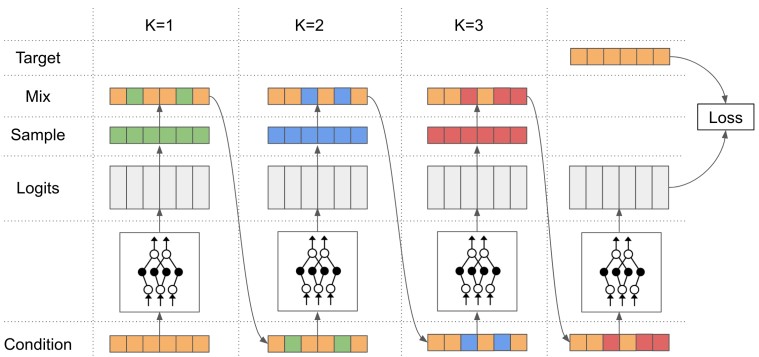

Figure 1: Parallel Scheduled Sampling for $K = 3$ passes. On each pass, the model is conditioned on the previous pass's output. New tokens are sampled according to the model's logits, which are then mixed with the gold sequence. After the final pass, the loss is calculated with respect to the gold tokens.

involving long sequence generation. In this work, we describe a simple technique to parallelize Scheduled Sampling. Given an input example, we first generate an entire model prediction sequence in parallel by conditioning on ground-truth history (equivalent to forward-pass of teacher-forcing). Then, we employ a (parallel) mixing step where we generate a new sequence whose token at every time step is either from the model prediction or the ground-truth. Finally, we perform training as in teacher-forcing by conditioning on the sequence obtained from mixing. In Section 2.3 we show that by performing multiple passes of parallel prediction and mixing, we obtain a conditioning sequence that converges to a sample decode from the model.

Our key contributions are,

- A novel, fully parallelizable approach to reducing train-test discrepancy of auto-regressive models without modification. We believe this to be the first approach that does not require full-sequence decoding within the training loop. We find the method produces the same empirical benefits of Scheduled Sampling while using as little as *0.3% of the training time*.

- We prove equivalence of the proposed approach to Scheduled Sampling under certain choices of hyperparameters. This recovers the clear interpolation between training-time teacher-forcing and test-time decoding described in Bengio et al. (2015).

- We extensively evaluate our approach on four auto-regressive tasks in text and image domains. We find that Parallel Scheduled Sampling matches Scheduled Sampling's benefits, takes massively less compute time, and significantly improves performance compared to teacher-forcing on most tasks. In dialog response generation task, Parallel Scheduled Sampling achieves 1.6 BLEU score (11.5%) improvement over teacher-forcing while in image generation it achieves 20% and 13.8% improvement in Frechet Inception Distance (FID) and Inception Score (IS) respectively.

## 2 METHOD

Our proposed technique can be applied to both conditional and unconditional auto-regressive generative models. For notational simplicity, we consider the task of conditional sequence generation. The training set is given in terms of $N$ input-output sequences $\{(x^i, y^i)\}_{i=1}^n$, where $x^i$ is the input and target $y^i$ is the desired output. The target $y^i$ is a variable-length sequence of $T_i$ tokens (or pixels), $(y_1^i, y_2^i, \ldots, y_{T_i}^i)$, whereas $x^i$ may be variable-length (as in translation) or fixed-length (as in image captioning). The goal is to learn a model that accurately predicts $y^i$ given $x^i$. We use $y_{1:t}$ to denote the sequence of tokens $(y_1, y_2, \ldots, y_t)$.

## 2.1 TEACHER-FORCING AND DECODING

Given an input $x$ and a target $y$, the log-probability of the target can be decomposed autoregressively:

$$P(y|x) = \prod_{t=1}^{T} P(y_t|y_{1:t-1}, x)$$

Auto-regressive sequence generation models learn to assign high likelihood to token $y_t$ given previous target tokens $y_{1:t-1} = (y_1, \ldots, y_{t-1})$ and inputs $x$ via a learned likelihood model $p_\theta$. Neural language models such as RNNs (Mikolov et al., 2010) and Transformer (Vaswani et al., 2017) adopt left-to-right decomposition while image generation models (van den Oord et al., 2016a; Parmar et al., 2018) adopt a raster-scan order.

Such models are typically trained with teacher-forcing (Williams and Zipser, 1989). In teacher-forcing, the log likelihood of the training set is directly maximized,

$$\theta^* = \operatorname*{argmax}_\theta \mathcal{L}_{\text{tf}}(\theta) = \operatorname*{argmax}_\theta \sum_{i=1}^{N} \sum_{t=1}^{T_i} \log p_\theta(y_t^i|y_{1:t-1}^i, x^i) \tag{1}$$

Importantly, teacher-forcing conditions on *gold* target prefixes $y_{1:t-1}$, enabling backpropagation through all timesteps with a single pass of inference.

At inference time, beam or sample decoding is often used to generate a candidate target $\hat{y}^i$. In this regime, target tokens are generated one at a time while conditioning on previously-generated tokens.

$$\hat{y}_t \sim p_\theta(\hat{y}_t|\hat{y}_{1:t-1}, x) \quad \text{or} \quad \hat{y}_t = \operatorname*{argmax}_y p_\theta(y|\hat{y}_{1:t-1}, x)$$

A potential failure mode for teacher-forcing-trained models is in conditioning on previously unobserved target prefixes $\hat{y}_{1:t-1}$. As the model has not conditioned on these prefixes at training time, it may generate bland, repetitive, or nonsensical candidate targets (Holtzman et al., 2019).

## 2.2 SCHEDULED SAMPLING

Scheduled Sampling (Bengio et al., 2015), hereafter Sequential Scheduled Sampling is a training technique designed to bridge the gap between teacher-forcing and sample decoding. In its simplest form, Sequential Scheduled Sampling generates tokens $\tilde{y}_{1:t}$ and conditions on these target prefixes during training. Sequential Scheduled Sampling uses the same objective function as teacher-forcing (Equation 1) except the conditioning tokens $\tilde{y}_{1:t}$ are a random mixture of gold tokens $y_{1:t}$ and sampled tokens $\hat{y}_{1:t}$ instead of gold tokens $y_{1:t}$. See Algorithm 1 for implementation.

---

**Algorithm 1** Sequential Scheduled Sampling (single example)

> **for all** timesteps $t = 1, \ldots, T_i$ **do**
>> Sample $\hat{y}_t \sim p_\theta(\hat{y}_t|\tilde{y}_{1:t-1}, x)$
>> Choose next conditioning token,
>>
>> $$\tilde{y}_t = \begin{cases} y_t & \text{with probability } 1-p \\ \hat{y}_t & \text{with probability } p \end{cases}$$
>
> **end for**
> **return** Accumulate loss $\sum_t \log p_\theta(y_t|\tilde{y}_{1:t-1}, x)$

---

As $p \to 0$, we condition on $y_{1:t-1}$ as in teacher-forcing, and as $p \to 1$, we condition on $\hat{y}_{t:t-1}$ as in sample decoding. Typically a schedule will be used to gradually increase $p$ over the course of training. As illustrated in Bengio et al. (2015), Scheduled Sampling leads to a performance improvement in a variety of language generation tasks.

In spite of its benefits, Sequential Scheduled Sampling is inherently a *sequential* algorithm: choosing conditioning token $\tilde{y}_t$ requires conditioning autoregressively on tokens $\tilde{y}_{1:t-1}$. While this is natural for sequential architectures such as RNNs and LSTMs, it is poorly suited to self-attending feed-forward models such as Transformer where inference for multiple timesteps can be carried out simultaneously.

## 2.3 PARALLEL SCHEDULED SAMPLING

We propose a natural extension to Sequential Scheduled Sampling called *Parallel Scheduled Sampling*. Whereas Sequential Scheduled Sampling selects conditioning tokens one after another, we propose generating conditioning tokens for all timesteps *in parallel* over the course of one or more passes. While this technique requires strictly more operations than Sequential Scheduled Sampling, it is better suited to hardware accelerators such as GPUs and TPUs (Jouppi et al., 2017). Moreover, we find in our experiments that only a modest number of passes is necessary for improving model performance.

Parallel Scheduled Sampling generates conditioning tokens for all timesteps *simultaneously*. The procedure consists of multiple passes, each pass consisting of parallel sampling and mixing steps (Figure 1). In the first pass, the algorithm conditions on gold tokens $y_{1:t}$, generating tokens $\hat{y}_{1:t}$ i.i.d. according to $p_\theta(\hat{y}_t|y_{1:t-1}, x)$. Sampling tokens in the first pass is equivalent to the forward-pass of teacher-forcing. The sampled tokens, $\hat{y}_{1:t}$, are mixed (in parallel) with gold tokens, $y_{1:t}$, to produce conditioning tokens for the next pass, $\tilde{y}_{1:t}$.

We now describe the multiple-pass procedure. Let $\hat{y}_{k,1:t}$, and $\tilde{y}_{k,1:t}$ denote sampled and mixed tokens respectively on pass $k$. The mixed tokens from pass $k$, $\tilde{y}_{k,1:t}$, are used for conditioning on pass $k + 1$ in place of gold tokens $y_{1:t}$. Finally, the loss is calculated as before, conditioning on the final mixture of gold and sampled tokens $\tilde{y}_{K,1:t}$. See Algorithm 2 for implementation.

---

**Algorithm 2** Parallel Scheduled Sampling (single example)

Set $\tilde{y}_{0,t} = y_t$.
**for all** passes $k = 1, \ldots, K$ **do**
    **for all** timesteps $t$ in parallel **do**
        Sample $\hat{y}_{k,t} \sim p_\theta(\hat{y}_{k,t}|\tilde{y}_{k-1,1:t-1}, x)$.
        **if** $t < k$ **then**
            Copy $\tilde{y}_{k,t} = \tilde{y}_{k-1,t}$
        **else**
            Sample $\tilde{y}_{k,t} = \begin{cases} y_t & \text{with probability } 1 - p \\ \hat{y}_{k,t} & \text{with probability } p \end{cases}$
        **end if**
    **end for**
**end for**
**return** Accumulate loss $\sum_t \log p_\theta(y_t|\tilde{y}_{K,1:t-1}, x)$

---

Finally, we prove that by running the sampling and mixing steps for multiple passes as described in Algorithm 2, the final sample from Parallel Scheduled Sampling converges to a random sample decode from the model when $p = 1$ and $K \geq T$.

**Theorem 2.1.** Consider a sequence of tokens $z = (z_1, z_2, \ldots, z_T)$ of length $T$. Let $p = 1$ and $K \geq T$ be fixed. Then the likelihood of $z_{1:T}$ under Parallel Scheduled Sampling's proposal distribution[1] over conditioning tokens on pass $K$, $q_\theta^K(z_{1:T})$, is identical to random sample decoding's, $p_\theta(z_{1:T})$, $q_\theta^K(z_{1:T}) = p_\theta(z_{1:T})$.

*Proof.* We begin by establishing notation. Let $p_\theta(z_{1:T})$ be the likelihood of a sequence $z_{1:T}$ according to random sample decoding. Let $q_\theta^K(z_{1:t})$ be the likelihood of the same according to Parallel Scheduled Sampling's proposal distribution on pass $K$.

The proof proceeds by induction. First we show that the proposal distribution for the first token matches random sampling's on the first pass, $q_\theta^1(z_1) = p_\theta(z_1)$. Then we show that if $q_\theta^K(z_{1:t}) = p_\theta(z_{1:t})$ holds for some $K$, it also holds for all $K' > K$. Finally, we show that if the previous

---

[1]We drop conditioning on $x$ in the following for conciseness

statement holds, it also holds for tokens $z_{1:t+1}$ on pass $K + 1$. Thus, it follows that the proposal distribution matches random sampling's for all $T$ tokens so long as $K \geq T$.

*Base Case*: Consider the proposal distribution for the first token on the first pass, $z_1$. As $p = 1$, the first token is sampled from $p_\theta(z_1)$ by construction. Thus,

$$q_\theta^1(z_1) = p_\theta(z_1)$$

*Induction over $K$*: Suppose that the proposal distribution for tokens $q_\theta^K(z_{1:t}) = p_\theta(z_{1:t})$ some $K \geq t$. Then the equality also hold for the proposal distribution on pass $K + 1$. This follows trivially as tokens $z_{1:t}$ are "copied" from pass $K$ to $K + 1$ and thus their likelihood is unchanged,

$$q_\theta^{K+1}(z_{1:t}) = q_\theta^K(z_{1:t}) = p_\theta(z_{1:t})$$

*Induction over $t$*: Suppose that the proposal distribution matches random sample decoding's for the first $t$ tokens for $t = K$; that is, $q_\theta^K(z_{1:t}) = p_\theta(z_{1:t})$. We show that the statement holds for pass $K + 1$ for tokens $z_{1:t+1}$. First, recall that by construction the proposal distribution for token $z_{t+1}$ given previous tokens $z_{1:t}$ is the same as random sampling's when $t \geq K$,

$$q_\theta^{K+1}(z_{t+1}|z_{1:t}) = p_\theta(z_{t+1}|z_{1:t})$$

Note that this only holds when $p = 1$. Then,

$$
\begin{aligned}
q_\theta^{K+1}(z_{1:t+1}) &= q_\theta^{K+1}(z_{t+1}|z_{1:t})q_\theta^{K+1}(z_{1:t}) \\
&= q_\theta^{K+1}(z_{t+1}|z_{1:t})q_\theta^K(z_{1:t}) \\
&= p_\theta(z_{t+1}|z_{1:t})p_\theta(z_{1:t}) \\
&= p_\theta(z_{1:t+1})
\end{aligned}
$$

Where we use the chain rule, induction over $K$ for $z_{1:t}$, the inductive assumption for $q_\theta^K(z_{1:t})$, and the definition of $q_\theta^{K+1}(z_{t+1}|z_{1:t})$ when $t \geq K$.

∎

## 3 RELATED WORK

A variety of approaches have been introduced since Bengio et al. (2015) to eliminate the train-test discrepancy also known as *exposure bias*. Goyal et al. (2017) integrate a differentiable approximation to argmax into the Scheduled Sampling procedure. Ranzato et al. (2015) propose MIXER, which finetunes a model trained with teacher-forcing with REINFORCE to maximize BLEU on a validation set. Zhang et al. (2019) propose a procedure similar to Scheduled Sampling but with an alternative method for selecting the model's next-token prediction. Both works inherently require full-sequence decoding within the training loop, making them impractical to apply in domains with long generations.

Concurrently, Mihaylova and Martins (2019) proposed a variant of Scheduled Sampling similar to the approach introduced in this work. In their work, only a single pass $K = 1$ is applied with the first pass conditioning on a mixture of input embeddings. Unlike this work, the proposed approach is not equivalent to Scheduled Sampling under any choice of hyperparameters. Lamb et al. (2016) introduce an alternative approach with similar motivation. A discriminator network is trained jointly with the generator to distinguish between generator's hidden states produced by conditioning on ground-truth and model prediction sample. The generator, apart from maximizing the likelihood of the data, is also trained to fool the discriminator (Goodfellow et al., 2014). With this new objective, the dynamics of the generator would be the same for conditioning on both ground-truth and model prediction. Our parallel sampling approach is orthogonal to professor forcing and can be potentially applied in their framework.

Another approach with similar motivation is proposed in Collins and Roark (2004). Rather than a maximum likelihood objective, the authors apply a variant of the Perceptron algorithm (Rosenblatt, 1958) which penalizes the model's most likely decode if it doesn't exactly match the target. In the time-series modeling domain, Venkatraman et al. (2015) propose DAD, wherein the training

set for a Markov dynamics model is augmented by pairing model predictions with ground-truth states. This method is similar in spirit to the approaches above except in its Markov assumptions. Further, methods with similar motivation have also been studied in sequential decision making and reinforcement learning setting (Daumé et al., 2009; Ross et al., 2011).

While our focus is on accelerating mitigation strategies for exposure bias, another line of work aims to accelerate sequential decoding. Stern et al. (2019) generate successively longer decodes by inserting tokens between pairs of already-decoded tokens. Roy et al. (2018) and Kaiser et al. (2018) generate tokens in parallel by additionally conditioning on a short sequence of discrete latent variables. Lee et al. (2018) apply an iterative refinement technique by performing multiple passes of parallel decoding.

## 4 EXPERIMENTS

We evaluate our proposed technique on image and text domains. In the text domain, we evaluate Parallel Scheduled Sampling on text summarization (Liu et al., 2018), task-oriented dialog response generation (Budzianowski et al., 2018), and machine translation (Sutskever et al., 2014; Vaswani et al., 2017) and compare it to teacher-forced training. We further evaluate the proposed technique on image generation on CIFAR-10. Since our procedure is intended to generate better sequences, we evaluate it at the sequence level and not at the word token or pixel level. We compare our method to Sequential Scheduled Sampling only on the dialog task (Budzianowski et al., 2018) as we find runtime infeasible on all other tasks. We use the Tensor2Tensor framework for all experiments (Vaswani et al., 2018).

### 4.1 DIALOG RESPONSE GENERATION

We evaluate our method on dialog response generation task using MultiWOZ (Budzianowski et al., 2018), a task-oriented dialog dataset. Here, we consider the problem of mapping conversation history consisting of alternating user and assistant turns to a single turn of assistant response. We use a Transformer model containing approximately one million parameters for this study as the dataset is much smaller (approximately 100k training examples) than those in other experiments. We truncate the length of the input and output to 512, and train all the models for 50k steps. As both model and dataset are small, we are able to empirically compare our method to Sequential Scheduled Sampling (such experiments are infeasible in larger models). Table 1 summarizes results for all experiments on the MultiWOZ dataset.

Both Sequential Scheduled Sampling and Parallel Scheduled Sampling (with just one pass) achieve better results than teacher-forced trained models. However, as can be seen in Table 1, Parallel Scheduled Sampling and teacher-forcing are both *two orders of magnitude* faster to train than Sequential Scheduled Sampling. A single pass of Parallel Scheduled Sampling is approximately 25% slower than teacher-forced training while producing the benefits of Sequential Scheduled Sampling. Table 1 also shows the impact of mixing probability, number of passes, warm-up steps, and the mixing probability schedule (Bengio et al., 2015) on model performance. Overall, we find a single pass with 50% gold/sampled mixing probability sufficient for improving performance. In the best setting, Parallel Scheduled Sampling achieves 1.6 BLEU score (11.5%) improvement over teacher-forcing.

### 4.2 SUMMARIZATION

Liu et al. (2018) propose a multi-document summarization task, where the task is to generate the text of a Wikipedia article given its references and other related documents. The dataset has close to 1.9 million training examples, and 230,000 test examples. We use a Transformer seq2seq model for this task in two hyper-parameter settings: a base model with 60 million parameters and a large model with 210 million parameters. For the base model, we restrict the maximum length of input and output to be 500, while for the large model the maximum length is set to 1500.

Table 2 shows the results of training base and large Transformer models for the summarization task. The base and large models were trained for 250k steps and 500k steps respectively. We use teacher-forcing for the first 50% of training steps in Parallel Scheduled Sampling as warm-up steps. The mixing probability is set to 50% and we perform a single pass of sampling and mixing (Algorithm 2).

| Training Method | Mixing Prob | Num Passes | Warm-up Steps | Schedule | Mean BLEU | Max BLEU | Training Steps/Sec |
|---|---|---|---|---|---|---|---|
| Teacher-Forcing | - | - | - | - | 14.11 | 14.62 | 47 |
| Sequential SS | 0.25 | - | 25k | exp | **14.35** | - | 0.13 |
| Sequential SS | 0.50 | - | 25k | exp | 13.84 | - | 0.13 |
| Sequential SS | 0.75 | - | 25k | exp | 12.97 | - | 0.13 |
| Sequential SS | 1.00 | - | 25k | exp | 3.95 | - | 0.13 |
| Parallel SS | 0.25 | 1 | 25k | exp | 14.13 | 14.50 | 35 |
| Parallel SS | 0.50 | 1 | 25k | exp | **14.55** | **14.74** | 35 |
| Parallel SS | 0.75 | 1 | 25k | exp | 14.06 | 14.32 | 35 |
| Parallel SS | 1.00 | 1 | 25k | exp | 5.63 | 6.24 | 35 |
| Parallel SS | 0.5 | 2 | 25k | exp | **14.60** | 14.75 | 27 |
| Parallel SS | 0.5 | 3 | 25k | exp | 14.32 | 14.73 | 23 |
| Parallel SS | 0.5 | 5 | 25k | exp | 14.33 | 14.70 | 17 |
| Parallel SS | 0.5 | 7 | 25k | exp | 14.56 | 15.07 | 14 |
| Parallel SS | 0.5 | 10 | 25k | exp | 14.55 | **15.21** | 10 |
| Parallel SS | 0.5 | 1 | 10k | exp | 14.24 | 14.88 | 35 |
| Parallel SS | 0.5 | 1 | 15k | exp | 14.56 | 14.98 | 35 |
| Parallel SS | 0.5 | 1 | 20k | exp | 14.52 | 15.01 | 35 |
| Parallel SS | 0.5 | 1 | 30k | exp | 14.66 | 14.98 | 35 |
| Parallel SS | 0.5 | 1 | 35k | exp | 14.56 | 15.11 | 35 |
| Parallel SS | 0.5 | 1 | 40k | exp | **14.73** | **15.38** | 35 |
| Parallel SS | 0.5 | 1 | 25k | linear | **14.49** | **14.76** | 35 |
| Parallel SS | 0.5 | 1 | 25k | sigmoid | 14.30 | 14.66 | 35 |

Table 1: Results from models trained with teacher-forcing, Sequential Scheduled Sampling, and Parallel Scheduled Sampling on dialog response generation. We report mean BLEU and maximum BLEU over 5 random restarts for each configuration except Sequential Scheduled Sampling, for which we report a single run. We provide results by varying different hyperparameters for both variants of Scheduled Sampling. We also provide training steps per second for the different training algorithms. In the best setting, Parallel Scheduled Sampling achieves 1.6 BLEU score (11.5%) improvement over teacher-forcing.

| Model Size | Max Length | Training Method | Decoding Method | ROUGE-2 | ROUGE-L |
|---|---|---|---|---|---|
| Base | 500 | Teacher-Forcing | Beam Search | 24.74 | 34.42 |
| Base | 500 | Parallel SS | Beam Search | **25.19** | **34.76** |
| Base | 500 | Teacher-Forcing | Greedy | 20.18 | 28.66 |
| Base | 500 | Parallel SS | Greedy | 22.09 | 31.16 |
| Large | 1500 | Teacher-Forcing | Beam Search | **30.98** | **39.83** |
| Large | 1500 | Parallel SS | Beam Search | 30.35 | 39.09 |
| Large | 1500 | Teacher-Forcing | Greedy | 29.25 | 37.91 |
| Large | 1500 | Parallel SS | Greedy | 29.42 | 38.35 |

Table 2: Performance on the summarization task using base and large Transformer when trained with teacher-forcing and Parallel Scheduled Sampling. We consider both beam search and greedy decoding. We adopt the widely-used ROUGE score as the evaluation metric (higher the better).

| Training Method | Mixing Prob | Num Passes | Warm-up Steps | Schedule | FID (↓) | IS (↑) |
|---|---|---|---|---|---|---|
| Ground Truth | - | - | - | - | 0.0 | 11.31 |
| Teacher-Forcing | - | - | - | - | 52.39 | 5.64 |
| Parallel SS | 0.10 | 1 | 100k | exp | 64.88 | 5.36 |
| Parallel SS | 0.25 | 1 | 100k | exp | 48.85 | 6.08 |
| Parallel SS | 0.50 | 1 | 100k | exp | **41.47** | 6.42 |
| Parallel SS | 0.75 | 1 | 100k | exp | 43.38 | **6.48** |
| Parallel SS | 1.00 | 1 | 100k | exp | 98.38 | 3.97 |

Table 3: Empirical results on CIFAR-10. We use Frechet Inception Distance (FID) (Heusel et al., 2017) and Inception Score (IS) (Salimans et al., 2016) metrics to evaluate the quality of the samples from teacher-forcing and Parallel Scheduled Sampling. We provide upper-bound scores by computing the metric on the ground-truth data.

With the base model, Parallel Scheduled Sampling obtains better performance than teacher-forcing with both beam search and greedy decoding while it performs better only with greedy decoding when the large model is used. Since we use models that are much bigger than the ones in the dialog task discussed before, it is runtime infeasible to apply Sequential Schedule Sampling here.

### 4.3 IMAGE GENERATION

In the image domain, we evaluate our method for image generation on the CIFAR-10 dataset. We compare class-conditional Image Transformer (Parmar et al., 2018) trained with teacher-forcing and Parallel Scheduled Sampling. After training, we decode a total of 50,000 randomly-sampled images conditioned on classes drawn from the training set. In additional to a baseline model, we compare all metrics to ground truth samples from the CIFAR-10 training set. We evaluate the image samples using Frechet Inception Distance (FID) (Heusel et al., 2017) and Inception Score (IS) (Salimans et al., 2016) metrics. These metrics have been widely used to evaluate the quality of image samples from GANs (Lucic et al., 2018; Miyato et al., 2018; Karras et al., 2018; Zhang et al., 2018).

Table 3 compares teacher-forcing with Parallel Scheduled Sampling on image generation. We train the 50 million parameter Image Transformer model for 200K steps in both the cases. We find that Parallel Scheduled Sampling with a single pass and a mixing probability of 50% significantly decreases FID by 20% and increases IS by 13.8% compared to the baseline. Similarly to our summarization experiment, it is runtime infeasible to apply Sequential Schedule Sampling here.

### 4.4 MACHINE TRANSLATION

We evaluate our method on the WMT 2014 English-German task which consists of approximately 4.5 million training sentences. We experiment with the large Transformer model that contains approximately 210 million parameters. We did not see performance improvements by using Parallel Scheduled Sampling. The model trained with teacher-forcing for 500k steps gets 28.74 BLEU. The same model trained with 250k warm-up steps using teacher-forcing and the next 250k steps trained with Parallel Scheduled Sampling with mixing probability set to 50% and a single pass of sampling and mixing (Algorithm 2) obtains 28.57 BLEU. Hyper-parameter tuning of warm-up steps and mixing probability did not improve performance. We hypothesize the lack of performance improvement may be due to the fact that the summarization, dialog response generation and image generation tasks have much longer output sequences than in machine translation, though further investigation is required.

## 5 CONCLUSION

We introduce a simple technique to parallelize Scheduled Sampling that allows Schedule Sampling to be applied for training models with hundreds of millions of parameters on large datasets. The technique potentially mitigates discrepancy between train and test time in auto-regressive sequence generation models. We find that in most cases our technique leads to better empirical performance on

summarization, dialog generation, and image generation compared to teacher-forced training. Our empirical results indicate that Parallel Scheduled Sampling can potentially improve the performance of auto-regressive sequence generation models particularly on tasks containing long sequences.

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
