# OpenReview forum: "Parallel Scheduled Sampling"
_ICLR.cc/2020/Conference — Reject_

### Official Review · AnonReviewer1 · 2019-10-23
**Official Blind Review #1**

**Rating:** 6

**Review:**

This paper proposes a technique for scheduled sampling that can be parallelized. At a high level, the method is to do teacher forcing, followed by passes of sampling from the model. During each round, the output of each pass are mixed with some probability. The time complexity then scales with the number of passes as opposed to the length of the sequence. The authors show results on dialogue generation, image generation, and machine translation. The proposed method generally obtains performance on par with scheduled sampling, but is much faster. The paper is well-written and the method clearly explained. I am not super excited about the technical contribution of this work, so my score is for weak acceptance.

Some feedback
- It would be nice to put the task in the caption for Table 1
- Figure 1 is underwhelming and doesn't really help illustrate how the method works, though its caption does.
- Because the main selling point of this is efficiency, I would like to see how it compares to work on non-autoregressive sequence generation (e.g. https://arxiv.org/pdf/1902.01370.pdf), which should probably be in the related works as well.

**Experience Assessment:**

I have published one or two papers in this area.

**Review Assessment: Checking Correctness Of Derivations And Theory:**

I assessed the sensibility of the derivations and theory.

**Review Assessment: Checking Correctness Of Experiments:**

I assessed the sensibility of the experiments.

**Review Assessment: Thoroughness In Paper Reading:**

I read the paper at least twice and used my best judgement in assessing the paper.

---

> ### Author Response · Authors · 2019-11-12
> **Thanks for the comments and the feedback!**
>
> We will add the task to the Table captions.
>
> Our work can be combined with the paper mentioned in the review (https://arxiv.org/pdf/1902.01370.pdf). Their work explores different generation orders (apart from left-to-right) while still suffering from the train-test discrepancy issue (exposure bias) discussed in our work. It would be interesting to apply Parallel Scheduled Sampling to train such models in future work.

---

> > ### Author Response · Authors · 2019-11-14
> > **Updated Draft**
> >
> > We've incorporated feedback from all reviewers into the latest revision of our draft.

---

### Official Review · AnonReviewer2 · 2019-10-23
**Official Blind Review #2**

**Rating:** 3

**Review:**

This paper proposes a parallelized version of Scheduled Sampling, which is proposed initially to mitigate Exposure Bias. Exposure Bias occurs when sequence generation models are trained by teacher-forcing, where the t-th token, $y_t$, is forced to be correctly estimated given ground truths, $y_1, ..., y_{t-1}$. Scheduled Sampling randomly selects the previous sequence from generated one or ground truth. Since sequential Scheduled Sampling requires a long time to train, the authors propose a way to parallelize it without degrading the performance. Experimental results for dialog response generation, summarization, and image generation (as a grammar model on images) demonstrate that the proposed method is faster than the original Scheduled Sampling. Additionally, the performance gain is shown in comparison to baselines trained by teacher-forcing.

My first concern is that several papers are not cited, although they also address Exposure Bias:
- Venkatraman et al., Improving multi-step prediction of learned time series models. AAAI, 2015.
- Ranzato et al., Sequence Level Training with Recurrent Neural Networks. ICLR, 2016.
- Zhang et al., Bridging the Gap between Training and Inference for Neural Machine Translation. ACL, 2019.
The second paper proposes MIXER, which is a method based on reinforcement learning. The training step can be performed without sequential sampling from the ground truth and predicted tokens. The third paper also randomly selects from a predicted sequence and the ground truth, but the selection is performed in a sentence-wise manner.

Related to the first concern, secondly, Scheduled Sampling itself is no longer the state-of-the-art method to solve Exposure Bias. It is unclear if the proposed method is competitive with the methods above. In addition, the empirical benefit in terms of training time seems to be small in comparison to them. The authors evaluated the proposed method using diverse tasks. Although it is good to see if Parallelized Scheduled Sampling is versatile, lack of comparisons to the existing methods except for the original Scheduled Sampling remains the superiority of Parallelized Scheduled Sampling unclear.

Finally, the technical contribution is not so fascinating. Although parallelizing the original Scheduled Sampling is practically essential, the proposed solution seems to be a straightforward way. Theoretical arguments and proof also look obvious.

Now I lean to reject this paper because of the concerns above. Since Exposure Bias is a fundamental problem for sequence generation problems as described in this paper, I would like the authors to revise their paper and submit it to another conference.

**Experience Assessment:**

I have published in this field for several years.

**Review Assessment: Checking Correctness Of Derivations And Theory:**

I assessed the sensibility of the derivations and theory.

**Review Assessment: Checking Correctness Of Experiments:**

I carefully checked the experiments.

**Review Assessment: Thoroughness In Paper Reading:**

I read the paper thoroughly.

---

> ### Author Response · Authors · 2019-11-13
> **Mentioned Prior Work requires full-sequence decoding in training loop**
>
> First and foremost, we would like to thank the reviewer for their thorough and insightful review of our submission. We particularly appreciate the highly relevant references to existing work that we were not aware of. We will include these references in a future revision of this submission.
>
> We would like to emphasize that the purpose of this work is to modernize an established technique for mitigating Exposure Bias. We introduce an approximation to Scheduled Sampling (Bengio et al, 2015) and validate that it not only provides equivalent performance to its predecessor but is also several orders of magnitude faster. We note that the prior works of (Ranzato et al, 2016) and (Zhang et al, 2019) both require full-sequence, autoregressive decoding **within the training loop**, which is precisely what makes (Bengio et al, 2015) infeasible for two of our three tasks. A direct comparison is simply not possible.
>
> Lastly, we argue that the simplicity of the proposed approach is a benefit rather than a drawback. The proposed method requires a single hyperparameter, is easy to implement, and shows practical improvements on the tasks considered. We believe this work will serve as an excellent baseline for future work to build on.

---

> > ### Author Response · Authors · 2019-11-14
> > **Updated Draft**
> >
> > We've incorporated feedback from all reviewers into the latest revision of our draft.

---

### Official Review · AnonReviewer3 · 2019-10-29
**Official Blind Review #3**

**Rating:** 6

**Review:**

Scheduled Sampling aims to overcome the problems that come with the discrepancy between train and test
when training sequence generation models using teacher-forcing. However one of the drawbacks scheduled sampling is that is it is hard to parallelize this is simply because some of the decode input tokens aren't known beforehand and need to be inferred i.e. loss of some tokens are dependent on some previous generations.
Authors introduce a method to perform parallel scheduled sampling by iteratively train using teacher forcing however after each iteration the gold references are mixed with the model predictions with a probability p, this allows parallelization as all decoder input tokens can be known beforehand. The final loss is calculated as before, conditioning on the final mixture.

In theorem 2.1 Authors proof that parallel scheduled sampling can converge to scheduled sampling when P is set to 1 and the number of iterations is larger than the sentence length.

In experiments over summarization, dialogue response generation and image generation authors show that the parallel scheduled sampling can achieve comparable performance to scheduled sampling with high-performance gains reaching 300 times faster and very comparable to teacher forcing.

pros:
The work presents a simple idea that is presented neatly with sufficient experiments. One of the outcomes of this method, that might not been stressed enough in the paper, is a neat way of doing scheduled sampling for the transformers architecture which wasn't straight forward before.
Current proposals include some architecture modifications such as in Mihaylova et al. 2019 https://arxiv.org/pdf/1906.07651.pdf

cons:
The only drawback I can find in this paper is it is lacking content. while the idea is interesting and supported by experiments, I find the content is slightly below the amount of content in average ICLR papers.

The explanation of the proposed scheduled sampling can be much simplified. The idea is quite simple however for example I find the pseudo code in Algorithm 2 is quite hard to grasp maybe due to some typos

Questions to authors:
- We only see performance comparison in dialogue response generation experiments. There are other factors that can affect performance or make parallelization effective.  I wonder what are the performance gains of parallel scheduled sampling on normal scheduled sampling with respect to avg. number of tokens / sentence or batch size.

- I had a hard time grasping Algorithm 2 although I understood the corresponding text could you please verify there aren't any typos. What it says is in the first iteration (k=1) scheduled sampling will be performed which doesn't make sense.


**Experience Assessment:**

I have published one or two papers in this area.

**Review Assessment: Checking Correctness Of Derivations And Theory:**

I assessed the sensibility of the derivations and theory.

**Review Assessment: Checking Correctness Of Experiments:**

I assessed the sensibility of the experiments.

**Review Assessment: Thoroughness In Paper Reading:**

I read the paper thoroughly.

---

> ### Author Response · Authors · 2019-11-12
> **Thanks for the comments and feedback!**
>
> We thank the reviewer for the comments on our submission.
>
> 1) Thanks for the pointer to the work on modifications to Transformer, we will cite and briefly compare to it in the next version of the manuscript.
>
> 2) The performance comparison experiment would be an interesting one. We could answer that question by training on sequences of difference length and compare the run time of normal and parallel scheduled sampling. We would report these numbers with the next version.
>
> 3) At k=1, **Parallel** Scheduled Sampling is performed and not the sequential one. The source of confusion might be due to the usage of the same symbol in both cases. We will fix this in the next version.

---

> > ### Author Response · Authors · 2019-11-14
> > **Updated Draft**
> >
> > We've incorporated feedback from all reviewers into the latest revision of our draft.

---

### Comment · Area_Chair1 · 2019-11-14
**Reviewers, any comments on the author responses?**

Dear Reviewers, thanks for your thoughtful input on this submission!  The authors have now responded to your comments.  Please be sure to go through their replies and revisions.  If you have additional feedback or questions, it would be great to get them this week while the authors still have the opportunity to respond/revise further.  Thanks!

---

### Decision · Program_Chairs · 2019-12-19

**Decision:**

Reject

**Comment:**

The paper proposes a parallelization approach for speeding up scheduled sampling, and show significant improvement over the original.  The approach is simple and a clear improvement over vanilla schedule sampling.  However, the reviewers point out that there are more recent methods to compare against or combine with, and that the paper is a bit thin on content and could have addressed this.  The proposed approach may well combine well with newer techniques, but I tend to agree that this should be tested.